# Status Quo in Data Availability and Predictive Models of Nano-Mixture Toxicity

**DOI:** 10.3390/nano11010124

**Published:** 2021-01-07

**Authors:** Tung X. Trinh, Jongwoon Kim

**Affiliations:** 1Chemical Safety Research Center, Korea Research Institute of Chemical Technology (KRICT), Daejeon 34114, Korea; trinhxt@krict.re.kr; 2Department of Chemistry, College of Natural Sciences, Hanyang University, Seoul 04763, Korea

**Keywords:** nano-mixture, toxicity, data curation, predictive models

## Abstract

Co-exposure of nanomaterials and chemicals can cause mixture toxicity effects to living organisms. Predictive models might help to reduce the intensive laboratory experiments required for determining the toxicity of the mixtures. Previously, concentration addition (CA), independent action (IA), and quantitative structure–activity relationship (QSAR)-based models were successfully applied to mixtures of organic chemicals. However, there were few studies concerning predictive models for toxicity of nano-mixtures before June 2020. Previous reviews provided comprehensive knowledge of computational models and mechanisms for chemical mixture toxicity. There is a gap in the reviewing of datasets and predictive models, which might cause obstacles in the toxicity assessment of nano-mixtures by using in silico approach. In this review, we collected 183 studies of nano-mixture toxicity and curated data to investigate the current data and model availability and gap and to derive research challenges to facilitate further experimental studies for data gap filling and the development of predictive models.

## 1. Introduction

The toxicity of single nanomaterials has been intensively tested in recent decades. Combinations of single nanomaterials and chemicals form nano-mixtures, which might cause co-exposure and mixture effects in living organisms. Because nanomaterials present different physicochemical properties in comparison with the properties of bulk chemicals, the mixture effects of nano-mixtures might deviate from the mixture effects of chemical mixtures (e.g., organic mixtures, metal mixtures, etc.). The in silico approach provides predictive models for reducing the experimental costs of in vitro/in vivo toxicity testing [1]. Experimental studies on the toxic effects of mixtures of different compounds have continuously increased, and conventional models, such as concentration addition (CA) and independent action (IA) models, have been frequently used to estimate the toxicity of mixtures based on the additive toxicity of single compounds [2,3,4]. Additionally, some quantitative structure–activity relationship (QSAR)-based models have been applied to mainly predict the toxicity of organic mixtures based on the structures of single compounds [5,6,7,8,9]. However, there have been few studies concerning the predictability of nano-mixture toxicity, and most of these studies employed the CA and IA models to test the applicability of the two conventional models [10,11,12,13,14,15,16,17,18]. The number of studies on QSARs for predicting nano-mixture toxicity is considerably lower, and thus far, there have been only three of them [10,11,12].

For chemical mixtures, previous reviews provided comprehensive knowledge on (1) computational models for chemical mixture toxicity [19,20] and (2) underlying mechanisms of mixture effects [21]. In order to apply in silico methods to nano-mixture toxicity, datasets, models, and toxicity mechanisms would be required. Current reviews for nano-mixture toxicity [22] focused only on a toxicity mechanism known as the Trojan horse phenomenon. There is a gap in the datasets and predictive models, which might cause obstacles in the toxicity assessment of nano-mixtures by using the in silico approach. To provide readers with a picture of the currently available datasets and predictive models of nano-mixture toxicity, in this review, we collected and curated 183 studies [10,11,13,14,15,16,17,18,23,24,25,26,27,28,29,30,31,32,33,34,35,36,37,38,39,40,41,42,43,44,45,46,47,48,49,50,51,52,53,54,55,56,57,58,59,60,61,62,63,64,65,66,67,68,69,70,71,72,73,74,75,76,77,78,79,80,81,82,83,84,85,86,87,88,89,90,91,92,93,94,95,96,97,98,99,100,101,102,103,104,105,106,107,108,109,110,111,112,113,114,115,116,117,118,119,120,121,122,123,124,125,126,127,128,129,130,131,132,133,134,135,136,137,138,139,140,141,142,143,144,145,146,147,148,149,150,151,152,153,154,155,156,157,158,159,160,161,162,163,164,165,166,167,168,169,170,171,172,173,174,175,176,177,178,179,180,181,182,183,184,185,186,187,188,189,190,191,192,193,194,195,196,197] of nano-mixture toxicity to investigate the current data and model availability and gap, and we derived research challenges to facilitate further experimental studies for data gap filling and the development of predictive models.

## 2. Literature Collection

Literature was searched from Web of Science (webofknowledge.com) and Google Scholar (scholar.google.com) using the keywords “mixture”, “toxicity”, and “nanomaterials”. Each publication was carefully checked and analyzed to appropriately gather the toxicity test datasets of mixtures of nanomaterials and other substances. Selected publications for the purpose of this review were studies assessing the co-exposure effects of nano-mixtures on in vitro models and environmental species such as crustaceans, fish, bacteria, plants, algae, and mollusks. The last search was conducted in June 2020. A total of 183 publications from 2005 to 2020 were sorted after the searching and checking process. Information on (1) nano-mixture composition, (2) toxicity tests, (3) toxicity endpoints, and (4) mixture toxicity models was extracted from the literature. Nano-mixture composition included the nanomaterials, mixed chemicals, and type of nano-mixture. Toxicity tests included the name and group of tested species and toxicity test guidelines. Toxicity endpoints are the names of the endpoints measured in the toxicity tests. Mixture toxicity models are the available models used to estimate or predict the toxicity of nano-mixtures.

## 3. Current Available Predictive Models for Nano-Mixture Toxicity

Conventional models for the joint toxicity of organic compounds include CA [2] and IA [4] models. Both models describe the additive joint toxicity of chemicals based on the similarity or dissimilarity of their mode of action (MoA). The models require the toxicity of single substances in a mixture to estimate the toxicity of the mixture as follows:(1)$$\mathrm{CA}\;\mathrm{model}:\;{\mathrm{ECx}}_{\mathrm{mix}}=\sum_{1}^{n}{\left(\frac{p_{i}}{{\mathrm{ECx}}_{i}}\right)}^{-1}$$
where ECx_mix_ is the predicted toxic effect of the mixture, *p_i_* is the fraction of component *i* in the mixture, and ECx*_i_* is the individual effect concentration when applied singly.
(2)$$\mathrm{IA}\;\mathrm{model}:\;Y=u_{\max}\prod_{1}^{n}q_{i}\left(C_{i}\right)$$
where Y denotes the biological response, *C_i_* is the concentration of chemical *i* in the mixture, *q_i_*(*C_i_*) is the probability of non-response, u_max_ is the control response for endpoints, and ∏ is the multiplication function. Several studies applied CA and IA models to predict the toxicity of nano-mixtures and showed good predictivity (correlation coefficient value of up to 0.91) and explain the synergism and antagonism effects of the mixtures [13,14,15,16,17,18]. For example, Lopes and colleagues [13] utilized CA and IA models for mixtures of ZnO/Ag nanoparticles (NPs) and their metal ion counterparts and showed that both Zn NP and Ag NP mixture showed a deviation from additivity and they showed synergism when the concentration of Ag ions increased and antagonism when the concentration of AgNPs increased in the suspension. The study by Wang and colleagues [15] demonstrated that toxicity predictions of CA/IA models were close to the observed joint toxicities of binary mixtures of graphene and ionic liquids. Although CA/IA models are able to provide hints about the synergistic/antagonistic effects of mixtures, experiments to determine the toxicity of all single components are needed in order to apply CA/IA models. Such a process requires high cost and time, and such experiments might not always be feasible.

QSAR models might be a helpful approach to compensate for the disadvantages of CA and IA models. QSAR models are mathematical relationships between endpoints (e.g., mortality, mitochondrial activity) and descriptors (e.g., concentration of mixtures, physicochemical properties of NPs). The QSAR model inputs do not require the toxicity of all single components in mixtures. Although there were 183 studies that investigated the toxicity of nano-mixtures, only three studies [10,11,12] developed QSAR models for the photocatalytic activity and toxicity of TiO_2_ NP-based nano-mixtures (Table 1). These studies were conducted to develop models for predicting the photocatalytic activity and cytotoxicity of nano-mixtures consisting of TiO_2_ NPs and (poly) metallic clusters (Au, Ag, Pd, and Pt). The nano-mixtures were heterogeneous photocatalytic NPs that could be used for ultraviolet and visible (UV–vis) light-induced processes to remove harmful pollutants from gas and aqueous phases. The two endpoints in these studies were the photocatalytic activity under UV–vis light of TiO_2_ nano-mixtures and the viability of Chinese hamster ovary (CHO-K1) cells exposed to TiO_2_ nano-mixtures. Descriptors of nano-mixtures (*D_mix_*) were calculated from quantum chemical descriptors of metallic clusters by using the additive equation:(3)$$D_{mix}=\%mol_{Me_{1}}\times D_{1}+\ldots +\%mol_{Me_{n}}\times D_{n}$$
where *D_mix_* is the additive descriptor of nano-mixtures, and $\%mol_{Me_{n}}$ and *D_n_* are the mole fraction and quantum chemical descriptors of metal cluster *n* in the mixtures. The additive equation was based on a simple additive approach for joint toxicity, where the properties of heterogeneous NPs are the result of the additive contribution of each component. Two algorithms were used to develop QSAR models: multiple linear regression (MLR) and decision tree (DT). These QSAR models showed good predictive power and successfully explained the observed photocatalytic activity and cytotoxicity of heterogeneous TiO_2_ NPs (over 90%). However, the data for model development were limited to 29 data rows of different metal cluster fractions so the applicability domain of these models was limited to only nano-mixtures of TiO_2_ NPs and four metal clusters (i.e., Ag, Au, Pt, and Pd). Furthermore, nano-toxicology regulation and assessment could adopt additional QSAR models of the toxicity of other NP-based nano-mixtures.

It is feasible to develop QSAR models which have a larger applicability domain for nano-mixture toxicity if larger datasets containing a variety of mixed chemicals are acquired. Our curated literature data contain studies of the in vivo/in vitro toxicity of NP–NP, NP–inorganic chemical, and NP–organic chemical mixtures (Figure 1A,B). TiO_2_, fullerenes (C_60_), multiwall carbon nanotubes (MWCNT), ZnO, and Ag are the most popular NPs in the tested nano-mixtures (Figure 1C). The curated data have been successfully exploited to develop QSAR models for single nanomaterial toxicity by using regression and classification algorithms [198,199,200,201,202]. QSAR model development for nano-mixture toxicity might apply similar data curation (e.g., data collection, data gap filling, normalization, etc.) and modeling methods (algorithms, validation) as in the case of single nanomaterials. However, descriptors for nano-mixtures (*D_mix_*) are very important and they are different from the descriptors of single nanomaterials. Because nano-mixtures consist of more than two components, their descriptors should contain information about all components (i.e., concentration, properties). Previous research suggested the calculation of *D_mix_* from the properties of all mixture components by using several methods [5,7,8,10,11,12]. The most used method is based on the assumption that components in a mixture act jointly by simple addition (Equation (3)). In addition to the additive D*_mix_*, other equations such as mean square and mean cubic [5,7,8,10,11,12] were previously applied to calculate *D_mix_* of organic mixtures, and models based on these mixture descriptors showed good prediction performance (R^2^ = 0.71–0.94). These equations and models have not been applied to nano-mixtures yet. Although there are several available methods for calculating *D_mix_*, they might not cover all nano-mixture toxicity data because the assumption of joint properties (e.g., additive) might not be suitable for all nano-mixtures. New methods for calculating *D_mix_* might be proposed during the development of QSAR models for nano-mixture toxicity. Because NPs and mixed chemicals are often different, in that one component is nano and the other(s) is bulk material, it is difficult to measure their common properties for *D_mix_* calculation. Instead of using experimental properties, theoretical properties are commonly used for nano-mixtures [10,11,12,203,204]. Simulation-based methods for theoretical properties can be classified as quantum mechanics (QM) or molecular dynamics (MD). Mikolajczyk and colleagues [10,11,12] applied QM simulation and density functional theory (DFT) calculation to small metal clusters (0.5 × 0.5 × 0.5 nm^3^) to obtain QM descriptors for *D_mix_* of heterogeneous TiO_2_ NPs. The *D_mix_* obtained from this QM approach (electronegativity) worked well to produce QSAR models that explained over 90% the observed photocatalytic activity and cytotoxicity of heterogeneous TiO_2_ NPs. Although the QM calculations could provide information on electronic energy (e.g., ionization, electronegativity, highest occupied/lowest unoccupied molecular orbital, etc.), calculations of large nano-sized systems are difficult and time-consuming, so this approach is limited by the size of NP clusters. The MD approach was utilized in the work of Burk and colleagues [203,204]. This approach simulated large Fe-Doped ZnO NPs (8–40 nm diameter) and *D_mix_* descriptors were based on a force-field calculation of the potential energies of whole NPs and the core and shell layers. The *D_mix_* descriptors obtained from this approach showed potential for linear regression (i.e., predicting cell viability, membrane damage, and mitochondrial reactive oxygen species (ROS) of HeLa and KLN205 cells with R^2^ = 0.740–0.877) and classification models (i.e., clustering toxicity data of Fe-Doped ZnO NPs by using principal component analysis with good accordance with the algal growth inhibition data) [204]. The MD method has a disadvantage in that it provides only topological and potential energy descriptors, which are not related to the electronic properties of nano-mixtures. In addition to the QM and MD approaches, the molecular descriptor approach might be applicable to nano-mixture toxicity data. The molecular descriptor method was previously used to obtain *D_mix_* for antibiotics and pesticide mixtures [204]. The QSAR models for antibiotics and pesticide mixture toxicity demonstrated high predictive performance (R^2^ = 0.9366). Their limitation is similar to the QM approach, namely that calculations for large-sized nano-mixtures are difficult and time-consuming. For QSAR model development of the current curated nano-mixture toxicity data, we suggest considering all three methods for finding the *D_mix_* with the best performance models and the most suitable toxicity mechanisms.

There are several algorithms for QSAR models suggested by the Organization for Economic Co-operation and Development (OECD) [205]: linear and nonlinear algorithms. For conventional organic mixtures, linear algorithms such as MLR are often used for developing models [5,6,7,8,9] because of their transparency and mechanistic interpretation. For nano-mixtures, only MLR and DT algorithms have been used for developing models [10,11,12]. Nonlinear algorithms such as random forest (RF) and artificial neural network (ANN) might be useful for developing models due to their high accuracy and robustness. However, transparency and mechanistic interpretation of these algorithms are lower than MLR and DT algorithms because they do not provide equations or single trees of prediction for other users who could use direct predictions (e.g., equations, etc.). The choice of algorithm is based on the model’s performance, transparency, and mechanistic interpretation. Therefore, it is time-consuming to choose the most suitable algorithms for QSARs of nano-mixture toxicity.

As a part of in silico research, nano-mixture toxicity QSAR models would provide low-cost toxicity screening of nano-mixtures, which would help to reduce animal testing and chemical waste [206]. Additionally, the models would provide warnings about potential nano-mixture toxicity when we design new nanomaterials (e.g., photocatalytic nanomaterials [10]) for safe-by-design approaches and minimal harmful effects. Another application of the models is checking the biocompatibility of new NP-based drug delivery. The NP-based drug might undergo many co-exposures to other organic chemicals and present mixture toxicity (e.g., AuNPs in presence of Polysorbate 20 synergistic toxicity at concentrations where the individual components were benign [52]). Finally, the models might help to control the processes of the release and emission of NPs into the environment, where flows of NP release might interact with dissolved organic matter and provoke mixture toxicity to the environment.

## 4. Current Available Data of Nano-Mixture Toxicity

Based on the types of mixed chemicals, nano-mixture toxicity data from 183 publications were divided into three groups: (1) NPs and NPs; (2) NPs and inorganic chemicals; and (3) NPs and organic chemicals (Figure 1A). The toxicity data were categorized into two groups: in vitro and in vivo (Figure 1B). The in vitro toxicity group included studies using bacteria, cells from animals, and humans as test systems (48 studies). The in vivo toxicity group included studies using animals as test systems (135 studies) (Figure 1B). The five most popular NPs in the tested nano-mixtures (i.e., TiO_2_, C_60_, MWCNT, ZnO, and Ag) occupied around 70% of the curated data (Figure 1C). Their data would be helpful for the meta-analysis of the toxicity of each NP-based nano-mixture.

We categorized the tested species into nine groups: crustaceans, fish, bacteria, plants, algae, cultured cell lines, mollusks, insects and worms, and fungi and others (Figure 2A). Among these nine groups, crustaceans, fish, and bacteria were the three most popular tested species, and 111/183 studies used these species to test the toxicity of nano-mixtures (Figure 2A). Metal oxide NPs were most frequently tested on these test systems, and 95/183 studies investigated the toxicity of metal oxide-based nano-mixtures (Figure 2A). Carbonaceous nanomaterial-based nano-mixtures were the second largest group, where 60/183 studies investigated their toxicity (Figure 2A). We further categorized 111 studies of crustaceans, fish, and bacteria into smaller groups of nanomaterials and species (Figure 2B–D). In the crustacean group, *Daphnia magna* (*D. magna*) was the most tested species, with 38/48 studies (Figure 2B). In the fish group, *Danio rerio* (*D. rerio*) was the most popular species, tested in 17/43 studies (Figure 2C). In the bacterial group, *Escherichia coli* (*E. coli*) was the most tested species, with 8/20 studies. TiO_2_ based nano-mixtures were most commonly tested for toxicity to crustaceans (13/48 studies) and fish (17/43 studies) (Figure 2B,C). C_60_-based nano-mixtures were the second largest nano-mixture group that was tested for toxicity to crustaceans (10/48 studies) and fish (5/43 studies) (Figure 2B,C). In the bacterial group, Ag NPs and ZnO NP-based nano-mixtures were the largest groups (10/20 studies).

Among the 183 studies, there were 88 different species and 48 different toxicity endpoints for all tested species. The endpoints of the three most popular species (*D. magna*, *D. rerio*, and *E. coli*) are shown in Figure 3. For *D. magna*, mortality and immobilization were the most abundant endpoints (45%), followed by bioaccumulation (23%), reproduction (8%), and oxidative stress (7%). Other toxicity endpoints (17%) for *D. magna* were uptake, metal ATPase activity, metallothionein inhibition, retention of dietary, cell damage, and hatching rate. For *D. rerio*, mortality was the most measured endpoint (19%), followed by bioaccumulation (17%), gene expression (15%), oxidative stress (10%), and hatching rate (8%). Other endpoints (31%) for *D. rerio* were cell viability, locomotion activity, malformation, abnormality rate, glutathione level, heartbeat, mitochondrial activity, thyroid hormone content, and uptake. For *E. coli*, growth inhibition was the most measured endpoint (40%). Other endpoints (60%) were oxidative stress, ATP production, photoproduction, indigo degradation, cell wall damage, cell viability, and bioaccumulation.

TiO_2_ NPs are often mixed with metal ions such as Cd^2+^, Cu^2+^, and As^5+^ and organic compounds such as pesticides and antibiotics for toxicity testing. C_60_ and MWCNTs are often mixed with organic compounds such as pesticides and antibiotics for toxicity testing. ZnO NPs are often mixed with nanometals/oxides and organic compounds for toxicity testing (Figure 1C).

In the 183 collected studies, nano-mixture properties were described by the properties of NPs and mixture components including core diameter, length (for carbon nanotubes), hydrodynamic diameter, surface charge, surface area, chemical composition, and crystallinity. If the mixture components were nanomaterials, then the core diameter was usually described. For inorganic/organic compounds, chemical names or formulas were provided.

As mentioned in the previous sections, the major test systems are *D. magna*, *D. rerio*, and *E. coli*. We summarize the data of these three species in Table 2, Table 3 and Table 4. Among the various nanomaterials tested for these species, TiO_2_ NPs, ZnO NPs, and C_60_ fullerenes were the most tested nanomaterials (Figure 2). According to OECD guidelines [205], to develop reliable QSAR models, defined endpoints should be considered. As shown in Figure 3, there are 34 endpoints for *D. magna*, *D. rerio*, and *E. coli*.

For *D. magna*, immobilization and mortality endpoints are defined based on two test guidelines, OECD 202 and ISO 6341, where the number of immobilized or dead *D. magna* is recorded and converted to a percentage over control samples. Because of their popularity in toxicity endpoints of *D. magna* (32 studies, 45% data)*,* immobilization and mortality datasets might provide a potential dataset with a large applicability domain for QSAR model development. In order to develop nano-mixture QSAR models to predict immobilization/mortality, one might start with a dataset containing only one NP, such as TiO_2_, ZnO, etc., to test the performance and mechanistic interpretation of models and then extend the nano-mixture dataset to other NPs and widen the applicability domain of the models. The bioaccumulation endpoint is the gradual accumulation of substances such as NPs, metal ions, or other compounds in an organism (e.g., *D. magna*, *D. rerio*, etc.). There are sixteen studies investigating the bioaccumulation of *D. magna* (Figure 3A) of six nanomaterials (TiO_2_, CeO_2_, Cu, MWCNT, C_60_, and SWCNTs), six inorganic compounds (Cu(NO_3_)_2_, CuCl_2_, CdCl_2_, ZnCl_2_, AgNO_3_, and Na_2_HAsO_4_), and five organic compounds (atrazine, methylparathione, pentachlorophenol, phenanthrene, and tributyltin). Previous studies suggested a mechanism by which higher bioaccumulation induced by the absorption of metal ions/dissolved organic matter and a low level of agglomeration might cause higher immobilization/mortality of *D. magna* [16,59,81,117,153,184]. For example, Martin-de-Lucia and colleagues [16] found that at low concentrations, the binary mixtures of graphite–diamond nanoparticles and fungicide thiabendazole expressed synergistic toxic interactions, which could be attributed to the increased bioavailability of fungicide thiabendazole adsorbed on graphite–diamond nanoparticles. At higher concentrations, because of agglomeration, the bioaccumulation decreased and so did the toxicity. Reproduction is another toxicity endpoint for *D. magna*, which follows the OECD 211 test guideline [207]. In the test guideline, the number of offspring produced by each parent animal is counted for the test and control samples. The reproduction dataset of *D. magna* might be a candidate for the development of QSAR models. Oxidative stress is the endpoint of measuring the production of reactive oxygen species (ROS). The ROS induction of nano-mixtures might be caused by the photocatalytic activity of Ag [14] or ion exchange and electrostatic adsorption to form surface complexes [181]. Data for each reactive oxygen species should be curated to obtain sufficient data to develop QSAR models.

For *D. rerio*, mortality is the most abundant endpoint in the current data collection (9 studies, 19% data). Endpoint measurement is based on OECD 236 test guidelines [208]; therefore, it is a well-defined endpoint for QSAR model development. There are only eight studies of *D. rerio* bioaccumulation (Figure 3B) for TiO_2_, C_60_, and eight organic/inorganic compounds. Bioaccumulation of nano-mixtures might be proportional to their toxicity to *D. rerio* [52,74,190] because of mixed chemicals (e.g., Polysorbate 20, etc.) assembled on the nanoparticle surfaces [52]. The hatching rate of *D. rerio* is an endpoint based on the OEDC 212 test guideline [209]; thus, it might be a well-defined endpoint for QSAR model development. Gene expression is a complicated endpoint due to various types of genes. Gene expression data are useful for establishing adverse outcome pathways (AOP) [210]. However, with seven studies investigating gene expression (Figure 3B) for *D. rerio*, well-defined toxicity endpoints relating to gene expression are important in QSAR model development and possible additional data need to be collected/produced in the future.

For *E. coli*, the current collected data only relate to fifteen studies including eight endpoints such as growth inhibition, ATP production, oxidative stress, and cell wall damage. The main data contain TiO_2_, ZnO, and Ag NP-based nano-mixtures. The data might be helpful for understanding the antibacterial effects of nano-mixtures and datasets for each endpoint could be exploited in QSAR model development.

## 5. Conclusions

In this review, we collected the current 183 studies (2005–2020) for nano-mixture toxicity and described current data and predictive models for nano-mixture toxicity. We found that, in the current data on nano-mixture toxicity, the *D. magna*, *D. rerio*, and *E. coli* datasets are the three datasets containing the most studies (38, 17, and 8, respectively). We suggest additional curation of these data for the development of QSAR models with respect to toxicity endpoints. In particular, based on a total of thirty-four toxicity endpoints of these three species, three specific datasets with well-defined endpoints would be potentially useful for QSAR model development: immobilization and mortality of *D. magna*; mortality of *D. rerio*, and growth inhibition of *E. coli*. Data for other endpoints might need further curation and additional experimental data to develop QSAR models. We also suggest potential descriptors for QSAR model development: mixture descriptors (*D_mix_*) could be calculated from descriptors of single components in mixtures obtained by quantum mechanics, molecular dynamics, and molecular descriptor approaches. The available formula of *D_mix_* based on the assumption of joint properties in mixtures might not always be suitable for nano-mixtures so a new formula for *D_mix_* might be required. A variety of linear and nonlinear algorithms, such as MLR, DT, RF, and ANN, might be used for developing QSAR models. The choice of algorithm depends on the nature of the datasets, predictive performance, and possible mechanistic interpretation. Future studies that apply these datasets for QSAR models of nanomaterial toxicity will be conducted to contribute to the risk assessment of nano-mixtures.

## Figures and Tables

**Figure 1 nanomaterials-11-00124-f001:**
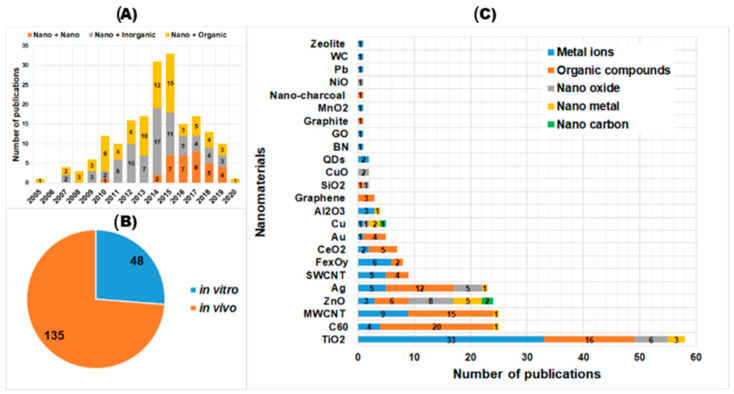
Number of studies on toxicity of nano-mixtures classified by year and type of nano-mixture (**A**), by type of toxicity test (**B**), and by nanomaterials and mixture components (**C**).

**Figure 2 nanomaterials-11-00124-f002:**
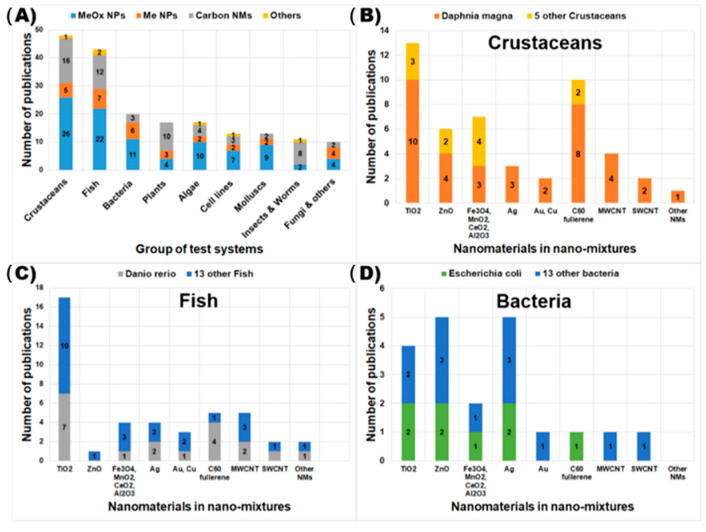
Number of studies on toxicity of nano-mixtures classified by group of test systems (**A**). Number of studies on toxicity of nano-mixtures classified by nanomaterials in nano-mixtures for crustaceans (**B**), fish (**C**), and bacteria (**D**). (SWCNT: single-walled carbon nanotube, MWCNT: multiple-walled carbon nanotube, NMs: nanomaterials).

**Figure 3 nanomaterials-11-00124-f003:**
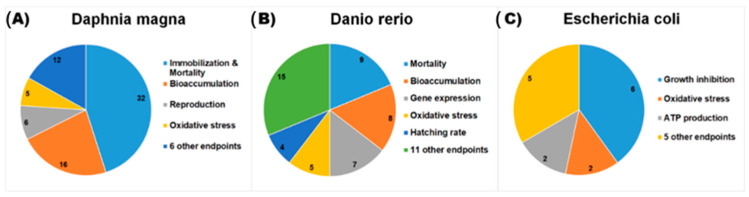
Number of studies on toxicity of nano-mixtures classified by endpoints for *D. magna* (**A**), *D. rerio* (**B**), and *E. coli* (**C**).

**Table 1 nanomaterials-11-00124-t001:** List of previous QSAR models for nano-mixtures.

No.	Reference	Nano-Mixture	Test System	Descriptors	Endpoint	Algorithm	Equations	No. Data	R^2^
1	Mikolajczyk et al., 2016 [12]	TiO_2_ NPs + Au/Pd	None	27 structural descriptors	Photocatalytic activity	Multiple linear regression	$\%\tau_{PhOH}=7.21+4.87\times XRD_{anatase}+2.44\times Pd_{\%mol}$	17	0.80–0.89
2	Mikolajczyk et al., 2018 [11]	TiO_2_ NPs + Au/Ag/Pt	Chinese hamster ovary cell line	5 quantum descriptors	Effective concentration (EC50)	Multiple linear regression	$\log{\left(EC_{50}\right)}^{-1}=4.69+0.18\times \%mol_{Ag}\;\log{\left(EC_{50}\right)}^{-1}=4.63+0.0004\times \mu_{mix}$	26	0.83–0.94
3	Mikolajczyk et al., 2019 [10]	TiO_2_ NPs + Au/Ag/Pt/Pd	Chinese hamster ovary cell line	9 quantum descriptors	Effective concentration (EC50)	Multiple linear regression	$pEC_{50}=6.37+0.56\times \chi_{mix}$	29	0.80–0.87
						Decision tree	None	29	0.74–0.90
					Photocatalytic activity	Decision tree	None	29	0.80–0.82

**Table 2 nanomaterials-11-00124-t002:** Summary of current nano-mixture toxicity data for *D. magna* (CA: Concentration Addition, IA: Independent Action).

No.	Article	Nanomaterials	Mixed Substance	Mixed Substance Type	Toxicity Test Guideline	Toxicity Endpoint	Mixture Toxicity Models
1	Azevedo et al., 2017 Science of the Total Environment [14]	ZnO NPs	Ag NPs	Nanometal	OECD 202; OECD 211	Immobilization, Reproduction	CA and IA
2	Baun et al., 2008 Aquatic Toxicology [98]	C60 fullerenes	Atrazine, Methylparathione, Pentachlorophenol, Phenanthrene	Organic compounds	OECD 211, ISO 6341	Reproduction, Immobilization, Bioaccumulation	None
3	Brausch et al., 2010 Environmental Toxicology and Chemistry [97]	C60 fullerenes	Bifenthrin, Tribufos	Organic compounds	EPA-821-R-02-012	Mortality, Reproduction, Growth inhibition	None
4	Cano et al., 2018 Environmental Science and Technology [178]	MWCNT	Cu^2+^	Metal ions	None	Metallothionein inhibition, Bioaccumulation	None
5	Fan et al., 2011 Environmental Pollution [107]	TiO_2_ NPs	Cu^2+^	Metal ions	None	Metallothionein inhibition, Mortality, Bioaccumulation	None
6	Fan et al., 2012 Journal of Nanomaterials [106]	TiO_2_ NPs	Cu^2+^	Metal ions	None	Oxidative stress, Metal-ATPase activity, Mortality	None
7	Fang et al., 2011 Environmental Toxicology and Chemistry [153]	Nano-charcoal	Tributyltin, Dibutyltin	Organometallic	ISO 6341	Reproduction, Immobilization, Bioaccumulation	None
8	Gao et al., 2018 Ecotoxicology and Environmental Safety [181]	MnO_2_ NPs, Nano-hydroxyapatite	Cd^2+^	Metal ions	OECD 202	Immobilization, Oxidative stress	None
9	Han et al., 2012 Chinese Journal of Geochemistry [158]	CeO_2_ NPs	Atrazine	Organic compounds	US. EPA 2001	Reproduction, Mortality, Bioaccumulation	None
10	Hartmann et al., 2012 Aquatic Toxicology [114]	TiO_2_ NPs	Cu^2+^	Metal ions	OECD 202, ISO 6341	Bioaccumulation	None
11	Kim et al., 2009 Environmental Science & Technology [136]	MWCNT	Cu^2+^	Metal ions	None	Bioaccumulation, Mortality, Oxidative stress	None
12	Kim et al., 2010 Environmental Toxicology and Chemistry [73]	Cu NPs	SWCNT	Nanocarbon	None	Mortality, Bioaccumulation	None
13	Lopes et al., 2016 J. Hazardous Materials [13]	Ag NPs, ZnO NPs	Ag NPs, ZnO NPs, Ag^+^, Zn^2+^	Nano-oxide, Metal ions	OECD 202	Immobilization	CA and IA
14	Martín-de-Lucía et al., 2019 Science of The Total Environment [16]	Graphite-diamond	Fungicide thiabendazole	Organic compounds	OECD 202	Immobilization	CA and IA
15	Molins-Delgado et al., 2016 Environmental Research [184]	Ag NPs	Benzophenone, Ethyl-PABA, 4-methylbenzylidene camphor, Ethylhexyl-methoxy cinamate	Organic compounds	ISO 6341	Immobilization	None
16	Park et al., 2019 Journal of Nanoparticle Research [117]	TiO_2_ NPs, ZnO NPs	Ag+	Metal ions	OECD 202	Immobilization	None
17	Park et al., 2019 Molecular & Cellular Toxicology [81]	Fe_3_O_4_ NPs	Zn^2+^	Metal ions	OECD 202	Immobilization	CA and IA
18	Rosenfeldt et al., 2014 Environmental Science & Technology [112]	TiO_2_ NPs	Cu^2+^, Ag^+^, As^5+^	Metal ions	OECD 202	Bioaccumulation, Immobilization	None
19	Rosenfeldt et al., 2015 Environmental Science & Technology [113]	TiO_2_ NPs	Cu^2+^	Metal ions	OECD 202	Bioaccumulation, Immobilization	None
20	Sanchis et al., 2016 Environmental Science & Technology [59]	C60 fullerenes	Nonylphenol, Triclosan, Malathion, Diuron, Glyphosate	Organic compounds	OECD 202, ISO 6341	Immobilization	None
21	Seitz et al., 2012 Environmental Toxicology and Chemistry [110]	TiO_2_ NPs	As^5+^	Metal ions	OECD 202	Immobilization	None
22	Tan and Wang, 2014 Environmental Pollution [109]	TiO_2_ NPs	Cd^2+^, Zn^2+^	Metal ions	None	Oxidative stress, Uptake, Retention of dietary, Mortality	None
23	Tan et al., 2012 Environmental Science & Technology [174]	TiO_2_ NPs	Cd^2+^, Zn^2+^	Metal ions	None	Uptake, Bioaccumulation, Retention of dietary	None
24	Tao et al., 2013 Chemosphere [193]	C60 fullerenes	Cu^2+^	Metal ions	US EPA 2024	Metal ATPase activity, Mortality, Bioaccumulation	None
25	Tian et al., 2014 Advanced Materials Research [108]	TiO_2_ NPs	Penta-brominated diphenyl ether	Organic compounds	OECD 202	Immobilization, Mortality	None
26	Vega et al., 2019 Ecotoxicology and Environmental Safety [65]	TiO_2_ NPs	Organic UV filter oxybenzone, Benzylparaben	Organic compounds	ISO 6341	Immobilization	None
27	Wang et al., 2014 Environmental Toxicology and Chemistry [148]	MWCNT	Ni^2+^	Metal ions	None	Immobilization, Bioaccumulation	None
28	Wang et al., 2016 Environmental Pollution [34]	MWCNT, SWCNT	Cd^2+^	Metal ions	OECD 202	Mortality, Bioaccumulation	None
29	Ye et al., 2018 Nanotoxicology [18]	ZnO NPs	GO NPs	Nanocarbon	OECD 201, 202, 236	Growth inhibition, Immobilization, Mortality, Oxidative stress	None
30	Yan et al., 2010 Chinese Science Bulletin [177]	C60 fullerenes	Atrazine	Organic compounds	None	Reproduction, Deformity rate, Hatching rate	None
31	Yang et al., 2010 Aquatic Toxicology [189]	C60 fullerenes	Fluoranthene	Organic compounds	EPA 6004-90027	Immobilization, Cell damage	None
32	Ye et al., 2018 Nanotoxicology [18]	ZnO NPs	GO NPs	Nanocarbon	OECD 201, 202, 236	Growth inhibition, Mortality, Immobilization	None
33	Yu & Wang, 2013 Water Research [197]	MWCNT, SWCNT	Cd^2+^, Zn^2+^	Metal ions	None	Uptake, Bioaccumulation	None
34	Yu & Wang, 2014 Environmental Toxicology and Chemistry [26]	C60 fullerenes	Cd^2+^, Zn^2+^	Metal ions	None	Bioaccumulation	None
35	Zhang et al., 2015 Journal of Environmental Sciences [31]	Ag NPs	Ag^+^	Metal ions	OECD 202	Mortality	None

**Table 3 nanomaterials-11-00124-t003:** Summary of current nano-mixture toxicity data for *D. rerio*.

No.	Article	Nanomaterials	Mixed Substance	Mixed Substance Type	Toxicity Test Guideline	Toxicity Endpoint	Mixture Toxicity Models
1	Azevedo Costa et al., 2012 Comparative Biochemistry and Physiology, Part C [176]	C60 fullerenes	As^3+^	Metal ions	None	Cell viability, Mitochondrial dehydrogenase functionality, Oxidative stress, Bioaccumulation	None
2	Fang et al., 2015 Journal of Hazardous Materials [160]	TiO_2_ NPs	Pentachlorophenol	Organic compounds	None	Bioaccumulation, Oxidative stress, Gene expression, Glutathione level	None
3	Ferreira et al., 2014 Aquatic Toxicology [131]	C60 fullerenes	Benzo[a]pyrene	Organic compounds	None	Cell viability, Oxidative stress, Bioaccumulation, Glutathione level	None
4	Ginzburg et al., 2018 ACS Nano [52]	Au NPs	Polysorbate 20	Organic compounds	None	Mortality	None
5	Henry et al., 2007 Environmental Health Perspectives [56]	C60 fullerenes	Tetrahydrofuran	Organic compounds	None	Mortality, Gene expression	None
6	Henry et al., 2013 Environmental Science & Technology [36]	C60 fullerenes	Hg+	Metal ions	Plymouth University	Mortality, Bioaccumulation, Gene expression	None
7	Hu et al., 2011 Environmental Pollution [38]	TiO_2_ NPs	Cd^2+^	Metal ions	None	Bioaccumulation	None
8	Hua et al., 2016 NanoImpact [74]	TiO_2_ NPs	ZnO NPs	Nano-oxide	OECD 157	Mortality	RA and CA
9	Krysanov & Demidova, 2012 Doklady Biological Sciences [146]	CeO_2_ NPs	Doxorubicin	Organic compounds	None	Malformations, Hatching rate	None
10	Miao, 2015 Aquatic Toxicology [119]	TiO_2_ NPs	Pb^2+^	Metal ions	None	Gene expression, Locomotion activity, Thyroid hormone content, Bioaccumulation	None
11	Park et al., 2011 Nanotoxicolog y [190]	C60 fullerenes	17α-Ethinyl-estradiol	Organic compounds	None	Bioaccumulation, Gene expression	None
12	Park et al., 2015 Molecular & Cellular Toxicology [60]	Ag NPs	Ag nanotube	Nanometal	None	Gene expression	None
13	Pavagadhi et al., 2014 Water Research [77]	Ag NPs, TiO_2_ NPs	Ni^2+^, Mg^2+^, Zn^2+^, Cu^2+^, Cd^2+^, Fe^2+^, Cr^3+^, Hg^2+^, As^5+^, Al^3+^, Pb^2+^, Mn^2+^	Metal ions	None	Mortality, Heartbeat, Hatching rate, Uptake	None
14	Wang et al., 2014 Nanotoxicology [84]	TiO_2_ NPs	Decabromdiphenyl ether	Organic compounds	None	Bioaccumulation, Locomotion activity, Oxidative stress, Gene expression	None
15	Ye et al., 2018 Nanotoxicology [18]	ZnO NPs	GO NPs	Nanocarbon	OECD 236	Mortality, Oxidative stress	None
16	Yan et al., 2014 Nanoscale Research Letters [103]	TiO_2_ NPs	Bis-Phenol A	Organic compounds	OECD 212	Hatching rate, Immobilization, Heart sac edema, Abnormality rate	None
17	Yan et al., 2018 Environmental Science and Pollution Research [166]	MWCNT	17 β -estradiol	Organic compounds	None	Mortality, Hatching rate, Abnormality rate	None
18	Zhang et al., 2012 Environmental Toxicology and Chemistry [140]	QDs (CdSe)	Cu^2+^	Metal ions	None	Mortality, Malformations, Hatching rate	None

**Table 4 nanomaterials-11-00124-t004:** Summary of current nano-mixture toxicity data for *E. coli*.

No.	Article	Nanomaterials	Mixed Substance	Mixed Substance Type	Toxicity Test Guideline	Toxicity Endpoint	Mixture Toxicity Models
1	Cuahtecontzi-Delint et al., 2013 International Journal of Chemical Reactor Engineering [138]	CeO_2_ NPs	Surfactants (Tween 80, Triton X114, and Polyvinyl Pyrrolidone)	Organic compounds	None	Growth inhibition	None
2	Li et al., 2005 Nanotechnology [134]	Ag NPs	Antibiotics	Organic compounds	None	Growth inhibition	None
3	Santaella et al., 2014 Environmental Science & Technology [55]	TiO_2_ NPs	Cd^2+^	Metal ions	None	Cell viability, Oxidative stress	None
4	Shahverdi et al., 2007 Nanomedicine Nanotechnology, Biology and Medicine [129]	Ag NPs	Antibiotics	Organic compounds	None	Growth inhibition	None
5	Silveira et al., 2015 Journal of Nanoparticle Research [147]	Ag NPs	Hexadecylpyridinium salicylate	Organic compounds	None	Growth inhibition	None
6	Srivastava et al., 2016 Journal of Environmental Sciences [133]	ZnO NPs	TiO_2_ NPs	Nano-oxide	None	Cell wall damage, Growth inhibition	None
7	Tong et al., 2015 Environmental Science & Technology [120]	ZnO NPs	TiO_2_ NPs	Nano-oxide	None	Photoproduction, ATP production, Oxidative stress, Bioaccumulation	None
8	Wilke et al., 2016 Environmental Science and Technology [57]	TiO_2_ NPs	Ag NPs	Nanometal	None	ATP production	None
9	Zhang et al., 2009 Environmental Science & Technology [37]	C60 fullerenes	Tetrahydrofuran (THF)	Organic compounds	None	Growth inhibition, Indigo degradation	None

## Data Availability

Data is contained within the article or Appendix A.

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
