# Peer review of "Status Quo in Data Availability and Predictive Models of Nano-Mixture Toxicity"

_nanomaterials, 2021, doi:10.3390/nano11010124_

Round 1

Reviewer 1 Report

The review by Trinh and Kim gives an overview of the published literature regarding mixture toxicity of different types of engineered nanomaterials or nanomaterials with other types of chemicals. They also survey the literature regarding descriptive and predictive models used in the mixture toxicity studies concerning nanomaterials. The data provided as figures and tables provide a useful overview of the state of the art in the mixture toxicity of nanoparticles and relevant modeling approaches. However, the text of the review seems to be lacking thorough analysis and critical review of the presented data. The authors mainly describe the results of the literature search and large parts of the text repeat the same information presented in the figures. The review would benefit from some additional in-depth discussion. Also, the suggestions for choosing toxicity endpoints for QSAR development do not seem well explained and justified. Some of the claims are just based on the lack of relevant toxicity data, but the data can always be created in the future experiments if it is deemed useful for modeling. The choice of data should be based on how well it predicts or describes NP toxicity and underlying mechanisms. Much of toxicity research, including nanotoxicity, is moving towards untargeted analysis of gene expression (transcriptomics), protein synthesis (proteomics) and metabolite levels (metabolomics). These techniques are believed to provide useful data for establishing adverse outcome pathways (AOP). In light of this overall development, the statement made by the authors “Gene expression is a complicated endpoint due to various types of genes; therefore, with only seven gene expression studies (Figure 3B), the data of each gene might not be sufficient for QSAR model development.” appears not justified and unclear. Similar unclear statements have been made about other toxicity endpoints in section 5. Structurally, it appears that section 3 (which currently mainly states the same information which is already evident from the figures) partially overlaps with the content of section 5. I would suggest to combine the contents of section 3 and section 5 to avoid repetition and unnecessary description of graphically presented information. Current section 4 could come right after section 2, because modelling is the main focus of the review and should be introduced to the reader first, before the literature data on organisms, toxicity endpoints and studied NPs and other compounds. After all, literature search was motivated by modeling approaches, thus, describing the findings after explaining the three modeling approaches seems logical. It seems that the authors have not sufficiently discussed the various co-contaminants of NPs reported in the literature. Since this is the main topic of the review it should receive more coverage, including adding references to the relevant sentences. It is unexpected that the reference list of a review paper only includes 18 items. At the same time there are clearly more papers cited in the tables of the review. These articles should also be listed as references in the end of the paper.

Reviewer 2 Report

The presented review is quite narrow in scope but the search was well executed. The text itself isn’t very long, which actually makes this relatively easy to follow.

 I would like the authors to consider whether the conclusions couldn’t be expanded to address applications more relevant to materials scientists. The review focuses on predictive models but a clearer message of what the consequences of this study might be for those who prepare the material would be very welcome.

Reviewer 3 Report

This is a very valuable research, because nanomaterials often do not exist alone. Nanomaterials often harm the environment and biology together with many chemicals. But some comments need to be addressed:

1, The author only gives some suggestions on the establishment of the model, but does not give the specific practice, feasibility and significance of these suggestions. The author should deeply discuss the idea and value of building model.

2, The authors consulted 183 references, but why only 18 references were cited? This is unfair and unscientific for most of the authors. 

3, The subscripts in most of the formulas in Tables 2 and 3 are wrong.

Reviewer 4 Report

The authors did a nice work. This is a systematic and comprehensive review paper, showing both views and facts. Thus, I suggest it be published after a minor modification.

Line 15: The authors mentioned the time was before mid-2020, but in line 51 they mentioned the last search was conducted in February 2020. Please consider unifying the date if it is necessary.

Table 2 and 3: Please unify the format of terms for mixed substances, such as the subscript.

Round 2

Reviewer 1 Report

The manuscript has been sufficiently revised and is now suitable for publication.

Reviewer 3 Report

Authors have addressed all comments well.